# Digital cultural intelligence and its role in enhancing expatriate work adjustment: A configurational approach in global work environments

Fatimah Mahdy◉*, Faiz Binzafrah, Osman Elsawy

Human Resources Management Department, College of Business, King Khalid University, Abha, Saudi Arabia

* fmhassan@kku.edu.sa

## Abstract

This study investigates how the digital dimensions of cultural intelligence influence expatriates' work adjustment across the Gulf Cooperation Council countries. Adopting a neo-configurational perspective, it explores four facets of digital cultural intelligence alongside contextual factors such as perceived organizational support, international experience, language proficiency, and digital-cultural training. Data from 208 expatriates working in tourism, healthcare, and higher education in Saudi Arabia, the UAE, Oman, and Qatar reveal multiple equifinal pathways to successful adjustment. Results highlight that adaptation emerges from distinct combinations of digital, cultural, and organizational factors rather than a single determinant. Two dominant configurations drive adjustment: high metacognitive and behavioral intelligence, combined with strong organizational support, and cognitive-motivational intelligence, paired with language skills and training to compensate for limited international experience. The study advances the person–environment fit theory by introducing the concept of digital–cultural fit and underscores the value of configurational methods in global HRM research.

## 1. Introduction

The last decade has seen the global labor market reshaped by rapid digitalization, integration, and globalization. Changing trends and patterns have affected the behavior of people and organizations when working or interacting in multicultural frameworks. Due to the spreading of remote and hybrid work models, professional environments have become even more complicated, with state-of-the-art technological instruments and a variety of cultural environments. Therefore, both players and workers need capabilities that go beyond technical know-how and that enable them

**Data availability statement:** All relevant data are within the manuscript and its Supporting Information files.

**Funding:** the Deanship of Research and Graduate Studies at King Khalid University, KSA, for funding this work through .(General Research Project under grant number (GRP/5/46) -2025. The funders had no role in study design, data collection and analysis, decision to publish, or preparation of the manuscript.

**Competing interests:** NO authors have competing interests.

**Abbreviations:** DCI: Digital Cultural Intelligence, CI: Cultural Intelligence, EWA: Expatriate Work Adjustment, POS: Perceived Organizational Support, IE: International Experience, LP: Language Proficiency, DCT: Digital Cultural Training, CT: Cultural Training, HRM: Human Resource Management

to successfully engage in virtual, cross-cultural contexts, defying anything as (cultural) blind spots, cognitive malleability, and adaptive capability [1,2].

In this line, the digital cultural intelligence (DCI) is seen as a further development of the traditional cultural intelligence (CI) concept [3] and it refers to the capacity of an individual to learn about, comprehend, and effectively behave in other cultural contexts. In today's digitally enabled workspaces, intelligence, unlike in the era that was dominated by face-to-face interactions as the modus operandi, is now mediated through technology. Thus, DCI is conceptualized as an individual's competence to apply cultural consciousness when using digital devices to enhance communication and collaboration across diverse cultural backgrounds within digital borders [4,5]. It is one of the most significant factors predicting performance in a multicultural digital work context.

While this remains the case, expatriate work adjustment (EWA) remains a vital construct in IHRM. Based on [6] seminal work, adjustment was considered a mental (cognitive processing of one's new environment), experiential (participating in and becoming socialized into the new environment), and sentiment-based (affective responses to the new surroundings) acculturation process that helps sojourners survive in culturally novel settings.

However, in more recent theories, adjustment has been redefined as a continuous learning process involving ongoing identity work and professional adjustment in situationally relevant conceptions of culture and organization [7,8], [9] highlight that digital workspaces compel expatriates to renegotiate their professional roles in ways that are increasingly temporal, cultural, and technologically complex, positioning work adjustment as more fluid and complex than ever. According to recent international human resource management (HRM) research, the results of international assignee adjustment suggest that it is not only a consequence of individual characteristics such as DCI but also of contextual and organizational aspects that exert a mediating/moderating influence [8].

They are now identified as perceived organizational support (POS), international experience (IE), language proficiency (LP), as well as digital and cultural training, which collectively influence the expatriates' capacity to convert CI into competent culturally adaptive actions. The significance of the study lies in its holistic view, which attempts to combine cultural and digital factors to predict individual outcomes within the global organization context.

This study advances the CI literature by re-positioning CI within a digital lens and simultaneously addressing contemporary challenges that expatriates face in virtual cross-cultural environments. This is particularly crucial for the Saudi and Gulf fields, where large-scale digital transformation and global talent mobility constitute the axes of Vision 2030. This study examines the influence of combined digitalization and human capital practices to enable sustainable competitiveness by integrating the concept of DCI in the process, allowing the adaptation of expatriates.

Drawing on Person-Environment Fit Theory [10], this study predicts that the fit between personal attributes (values, skills, and knowledge) and the demands of the environment will be positively associated with job satisfaction and job performance.

In today's digital workplace, such alignment transcends spatial or cultural parameters and embraces the notion of 'digital–cultural' fit, that is, an individual's capacity to digitally inscribe the cultural self within the technological work environment [11].

Therefore, DCI functions as a mode of contemporary fitness in winter, enabling employees to align themselves and the medium through reading, interpreting, and participating via culturally anchored digital signifiers [1].

To overcome the weaknesses of linear models in reflecting emergent properties in complex sociotechnical systems at the organizational level, this study adopts a Configurational Perspective [12], which posits that organizational outcomes are the result of the interaction effects among multiple elements rather than the outcome of the isolated effects of each element.

More specifically, using a fuzzy-set Qualitative Comparative Analysis (fsQCA), it analyzes the positive and negative causal relations that contribute to expatriate positive adjustment and predicts the combination of DCI, POS, and IE that produces the highest success. The concept of configuration allows us to account for multiple equifinal paths that result in the accomplishment of the same outcome, which is essentially what we have to consider with the fluid and nonlinear nature of global digital work sites.

Therefore, the central research question guiding this study is: To what extent does DCI enhance EWA in virtual multicultural environments?

1. Which features of the DCI predict expatriates' work adjustment?

2. Is the impact of the DCI the same at both ends of IE and POS?

3. How well can fsQCA be used to detect the "best" combinations of the factors in work adjustment prediction?

The above-mentioned motivations for the investigation are as follows:

• To evaluate expatriates' work adjustment in terms of the four DCI subscales.

• This study investigates the moderating and mediating roles of POS and IE in this relationship.

• To formulate a configurational theory-based model that describes different causal paths for achieving successful expatriate adjustment.

Theoretically, this study makes two contributions. First, it redefines CI in the digital environment and contributes to the literature on how people use technology and culture simultaneously. Second, it integrates DCI into the Person–Environment (P–E) Fit framework and proposes a multilayered dynamic among the individual, the organization, and culture. Moreover, this research contributes to the application of fsQCA in HRM studies by revealing novel, more fine-grained insights into how, when, and to what extent complex causality plays out as contextual equivalence.

Practically, the expected findings may offer strategic and operational implications for efficiently managing global talent in a digitalized world. They may inform the design of training interventions to enhance DCI and organizational support systems that integrate cultural and technological dimensions. This would be particularly beneficial for organizations in Saudi Arabia and the Gulf, which are embracing digital transformation as a component of their business strategy in the wake of Vision 2030 to boost global workforce readiness.

Ultimately, this study adds to contemporary discussions by moving beyond the "technology vs. culture" dichotomy and by providing an integrative framework that demonstrates that what makes the twenty-first-century successful sojourner is predicated not simply on IE but on having DCI the ability to turn cultural plurality into a source of innovation and sustainable competitive advantage.

The Gulf Cooperation Council (GCC) countries represent a theoretically distinctive context for examining expatriate work adjustment (EWA) and digital–cultural integration (DCI). Unlike traditional expatriation settings in Western economies, GCC labor markets are characterized by a pronounced reliance on expatriate labor, often exceeding national

workforce participation in key sectors such as healthcare, tourism, higher education, and advanced services [13,14]. Expatriates in the GCC typically work within highly regulated employment systems, short- to medium-term contractual arrangements, and institutionally constrained mobility structures, all of which significantly shape the adjustment process.

Expatriation in the GCC is influenced more by contextual fit and functional adaptation than by long-term assimilation, due to environments constrained by cultural, legal, and digital factors. The temporary nature of these assignments increases the importance of rapid adjustment mechanisms, particularly those enabled by digital communication, virtual collaboration, and technologically mediated social interaction [9]. Consequently, expatriate success depends less on gradual cultural immersion and more on the ability to navigate culturally embedded norms through digital platforms and organizational systems.

GCC countries exhibit strong institutional and cultural continuity while simultaneously pursuing ambitious national digital transformation agendas, as demonstrated by initiatives such as [15] and comparable strategies across the region. The coexistence of digital acceleration and institutional rigidity creates a distinctive tension for expatriates, who must reconcile technologically enabled work practices with deeply rooted cultural expectations and governance structures [11,14,16].

These characteristics position the GCC as a valuable context for extending Person–Environment Fit theory beyond traditional cultural congruence to include a broader concept of digital–cultural fit. Within this framework, successful adjustment arises from the alignment between individual digital-cultural competencies and institutionally constrained work environments. Accordingly, research on EWA in the GCC provides both context-specific insights and advances international human resource management (HRM) theory by illustrating how DCI operates under significant institutional constraints and digital mediation.

While this study is situated within the GCC context, it does not seek to offer universal recommendations for global human resource management practice. Instead, the GCC serves as a theoretically informative case that facilitates a deeper understanding of EWA under conditions of strong institutional regulation, significant expatriate dependence, and digitally mediated organizational interaction. The primary contribution of this study is to advance theoretical insights rather than to propose context-independent managerial solutions.

## 2. Theoretical and conceptual framework

### 2.1. Digital cultural intelligence (DCI)

The last two decades have witnessed a rapid pace of digitalization with complexity and cultural diversity in all layers of organizations and educational environments [1]. This is similar to the inadequacies of being only technically competent while needing cognitive and behavioral capabilities that enable one to participate with CI in digital contexts that are not housed in material borders.

In this context, CI has been first proposed by [3] and it is now considered the most solid theoretical framework for explaining people behavior in globalized, digital environments [1]. It combines knowledge, metacognition, motivation, and behavior in a way that allows individuals to think about, be motivated to, and act on cultural differences [17].

In digital age, CI may be viewed as a cognitive–affective–behavioral entity that individuals utilize digital-related competencies to engage in human-centered interactions and to become effective in performing within culture-specific, technological virtual context [18]. CI is the ability of a person to recognize and make meaning of cultural differences in behaviors and to respond appropriately when interacting with people from other cultures [3] and is conceptualized as an individual's social intelligence within the realm of culture.

Over the past two decades, this notion has been further developed to incorporate digital mediation as a feature of intercultural communication methods. CI is also expanding from the traditional face-to-face interactional context to include online communication, virtual teams, and borderless learning environments [1,2]. The outbreak of COVID-19 stimulated this need, giving rise to the concept of DCI, which is the capacity to utilize technology in a culturally sensitive way that fosters understanding and collaboration among diverse individuals [4,5].

                                                                   

CI frameworks were previously centered on the interpersonal domain and active behaviors; however, CI is increasingly being conceptualized as having a cultural aspect in digital practices, such as in virtual environments, multicultural teams via digital instruments, and interculturally tailored digital content [4,5].

Hence, DCI is a cognitive extension of emotional and social intelligence theories (LC) [3] with a metacognitive-reflective aspect in which individuals construct meaning about their cognitive schemas during rather than post (or 'synchronously') intercultural online communications courses and not solely reactively engage in processing of cognitive schemas. Early models did not pay sufficient attention to the convergence of culture and technology [18]. The new perspective of Digital Humanism fills this gap, treating technology as a tool for enhancing cultural awareness and aligning it with human values.

Therefore, DCI goes beyond just being culturally adaptable to become a strategic capability for managing digital differences—deciding how values are enacted through data-driven, AI-mediated spaces. In this sense, to create a thriving digital ecosystem, one must balance technology with culture, with technology offering the tools, and DCI the consciousness and behavioral ability to apply those tools in meaningful ways.

**2.1.1. Multidimensional structure of DCI.** Conceptual unification [17]. Currently, four integrated aspects of CI are becoming even more relevant in the digital age:

**Metacognitive CI:** Metacognitive CI is an individual's ability to be aware of and reflect critically on one's own intercultural interactions. It is a reflective stance that perpetually questions its own cultural premises, even as it is operating. As for the digital world, this facet is represented by consciousness toward digital signs and virtual subcultures – such as online writing culture, identity through visual codes, culture familiarity, or interaction cultures.

Metacognitive DCI decay (intercultural digital communication strategy) a set of leaders who successfully alter and adapt their communication platform – specific and cultural-based strategies rules of communication govern in their minds [2]. The skill is the basis of Digital Metacognition, a newly developed indicator of professional excellence in virtual workplace. Accordingly, the hypothesis can be formulated as follows:

**H2a: Reflective cultural awareness in virtual interactions mediates the positive impact of metacognitive CI on expatriates' overall adjustment**. *Cognitive CI*

Cognitive CI refers to the extent to which people possess knowledge pertaining to cultural values and behaviors. Cognitive CI illustrates how much an individual knows about the values, norms, and general customs of a particular country or organization [3]. In the liquid work regimes, it is not just about cultural knowledge because digital systems are cultural carriers (the systems embody certain specific cultural values within their structures and use in forms of packages and applications).

According to [1], knowledge of digital communication protocols, such as time-zone awareness, virtual meeting etiquette, and cross-platform coordination, is becoming digital cultural literacy that mitigates miscommunication and supports leadership effectiveness in international teams. Hence, the hypothesis is proposed as follows:

**H2b: Cognitive CI positively influences work adjustment by enhancing expatriates' knowledge of host organizational and digital work cultures.**

**Motivational CI:** Motivational CI is the interest/motivation to engage in multi-racial cultural dominance. Motivational CI is the innate interest and motivation to engage with culturally different individuals and situations. It is also a form of psychological energy that promotes the smooth flow of digital pluralism. [19] show that a strong motivational CI buffers against an ethnocentric attitude and enhances knowledge sharing in heterogeneous organizations. In a digital learning environment [5], associate motivational CI with learners' engagement and intention to engage with content containing local wisdom, which may cultivate learners' emotional attachment to and belonging to their culture. Therefore, the hypothesis may be stated as follows:

 

**H2c: The positive relationship between motivational CI and interaction adaptation is partially due to expatriates' increased openness and desire to communicate across digital-cultural borders.**

**Behavioral CI:** Behavioral CI "refers to the capacity to demonstrate appropriate verbal and nonverbal behaviors (such as the pace in conversation and body movements) in intercultural interactions" [17]. In digital workspaces, this also applies to (virtual) communication behavior (e.g., choice of channel, timing of response, style and content of writing, use of emojis and privacy management).

[4] signal this dimension as a shift from cultural cognition to digital ethos – that is, to transforming cultural understanding into culturally competent behavior online. Based on the above, the hypothesis can be articulated as follows:

**H2d: Behavioral CI is expected to have a positive influence on general and interaction adjustment by enabling expatriates to more easily modify their digital communication behavior in a foreign culture.**

These items, taken together, represent an integrated digital adjustment process and imply that higher levels of DCI will be positively correlated with higher levels of cognitive, affective, and behavioral adjustment among expatriates.

### 2.2. Intra-cultural heterogeneity and divergent digital–cultural fit in the GCC

A distinctive feature of human resource management in the GCC context lies not primarily in the coexistence of multiple national cultures, but in the markedly heterogeneous behavioral responses exhibited by individuals from similar cultural or demographic backgrounds. Expatriates originating from comparable home cultures often experience divergent trajectories of adjustment, engagement, and perceived cultural fit within the same organizational and institutional environments. This phenomenon is particularly salient in the GCC, where expatriates operate under shared regulatory regimes, standardized contractual arrangements, and relatively homogeneous organizational governance structures.

Segmented assimilation theory provides a useful lens for understanding this divergence, suggesting that individuals exposed to similar structural conditions may nonetheless follow differentiated paths of integration depending on their access to resources, relational positioning, and strategic responses to institutional constraints [20,21]. Applied to the GCC labor market, this perspective implies that expatriates with similar cultural origins do not converge toward a single mode of adaptation but instead develop distinct forms of digital–cultural fit shaped by their interaction with organizational norms, digital work practices, and social environments.

Social network theory further illuminates this process by emphasizing how patterns of relational embeddedness condition adjustment outcomes. Acculturational homophily suggests that individuals tend to form ties with others who share similar backgrounds; however, the extent to which expatriates rely on homophiles versus heterophilies networks can produce markedly different adjustment experiences [22]. In digitally mediated workplaces, these dynamics are intensified as professional visibility, collaboration, and inclusion are increasingly shaped by virtual interaction and platform-based communication.

Consequently, two expatriates with similar cultural or demographic profiles may diverge significantly in their adjustment outcomes: one may achieve alignment through strong organizational networks and adaptive digital behaviors, while another may experience marginalization despite cultural similarity. This intra-cultural heterogeneity challenges universalistic HRM assumptions and underscores the need for more differentiated expatriate management strategies. From a policy and HRM perspective, the findings suggest that effective governance of expatriate labor in the GCC requires moving beyond standardized cultural training (CT) toward more flexible, network-aware, and digitally attuned HR practices that recognize multiple viable pathways to adjustment and inclusion.

### 2.3. Expatriate work adjustment (EWA)

EWA is now considered a foundational notion in the fields of international HRM and global OB. In the context of globalization and digitalization that are reshaping how work is organized, adjustment is no longer viewed as a transitional process but rather as a strategic resource underpinning professional performance and well-being [8,23].

The classical model by Black, Mendenhall and Oddou (1991) considered adjustment as a one-dimensional, gradual process of cognitive, behavioral, and emotional convergence. New perspectives, however, view it as reflecting an ongoing learning cycle, through re-interpretation and re-definition of professional meaning and identity as individuals encounter new cultural and institutional conditions [7].

Thus, work adjustment for expatriates is a two-way interaction between the person and the environment and is constantly evolving through personal and POS factors. In the virtual world, expats also need to redefine their professional roles in line with remote, multicultural, and technology-mediated conditions. Similarly, adjustment is described as a cyclical process that extends beyond the timeframe of the assignment and encompasses the phases of repatriation and reintegration [24].

While the model of [6] differentiated among three levels of general, work, and interaction adjustment, it has been argued that it is too static. Modern perspectives [25] treat adjustment as a processual mode of being affected by ongoing interaction between the person and the context. [7] conceptualize adjustment as deep organizational learning/identity transformation, and is consistent with the Configurational Theory of Hinings (1993), which argues for non-linear causation.

Recent literature, including three key points of focus:

- Cognitive lets the individual understand the host country practices, the culture of the organization as well as the professional conduct expected [25]. Behavioral Adaptation: The ability to adapt one's behavior and communication, including virtual collaboration and digital communication styles, to the host cultural norms [11,26].

- Emotional Adjustment: The individual's emotional adaptation to the new surroundings [27].

Therefore, adjustment is a dynamic interplay of individual, organizational, and contextual factors. At the micro level, cultural intelligence, cognitive flexibility and self- motivation are the major constructs [8]. POS, mentoring, and leadership style contribute to overall integration success on the organizational level as well [26,28]. Finally, even adjustment procedures are further mediated by contextual factors, such as societal and technological factors [9].

Therefore, it is hypothesized that the four dimensions of DCI have a combined positive effect on the three adjustment dimensions for expatriates:

H2: The four dimensions of DCI (Metacognitive, Cognitive, Motivational and Behavioral) have a positive effect on the 3 dimensions of EWA(General, Work and Interaction).

## 2.4. Mediators and moderators in the proposed model

This study adopts a multilevel perspective, considering individual, organizational, and environmental factors to explain the DCI–EWA relationship. Beyond the direct impact, many influences mediate and moderate the efficacy of DCI intelligence [1,12].

### 2.4.1. Perceived organizational support (POS) as a mediator.
POS—employees' general belief about the degree of care between an organization and the individual [29] is the core psychological process linking DCI to adaptation outcomes. A high level of perceived support enables expatriates to use DCI to solve work and technical issues, which then results in a good work adjustment [10]. From this point, the hypothesis can be constructed as follows:

**H3: The positive relationship between DCI and EWA is mediated by POS.**

International Experience (IE): IE denotes exposure to other cultures. Reviewing previous research, novice expatriates are considered more dependent on DCI to compensate for their lack of knowledge of the host country environment, while experienced expatriates are considered to rely more on conventional adjustment means [7]. Hence, the hypothesis is proposed as follows:

**H4: The negative effect of DCI on the EWA of less-experienced expatriates becomes less negative as their IE increases.**

**Language Proficiency (LP)**: Linguistic competence encourages intercultural communication at both the linguistic level. [9] contend that the influence of DCI on interaction adjustment is enhanced by greater linguistic proficiency, which enables more accurate interpretation of cultural signs in virtual environments. H5: The relationship between DCI and Interaction Adjustment is positively moderated by LP.

**2.4.2. Digital and cultural training (DCT).** DCT is a strategic development lever that advances both the technical and the cultural dimensions of online teamwork. DCT is a unique form of strategic developmental leverage that simultaneously strengthens both the technical and cultural dimensions of online teamwork. Studies have demonstrated that techno-CT can significantly improve a person's capacity to engage with cultural differences in the digital realm [5]. Accordingly, the hypothesis can be formulated as follows:

**H6: The positive relation between DCI and EWA is strengthened by Digital and Cultural Training.**

**2.4.3. Relationships in the configuration.** From a configurational lens, this study posits that the success of expatriates depends on a constellation of antecedents rather than a single antecedent. As [12] noted, there are "multiple causal paths to good outcomes" (equifinality). To identify different causal recipes for optimal adjustment (e.g., high DCI with high POS or in combination with intense training), the study uses fsQCA. Therefore, the hypothesis may be stated as follows:

**H7: There are multiple causal patterns of DCI, POS, IE, LP, and DCT that predict successful work adjustment for expatriates.**

**H8: The synergistic impact of DCI with contextual factors (support, experience, language, and training) forms a whole that positively influences the work adjustment of expatriates.**

Based on the previous assumptions, it can be explained in the following conceptual model, Fig 1:

## 3. Methods

### 3.1. Sample description and data distribution

The final sample comprised 208 completed surveys from practitioners in four GCC states: Saudi Arabia (39%), the United Arab Emirates (28%), Oman (19%), and Qatar (14%). The respondents were from the following three sectors: tourism and hospitality (n = 45; 21.6%), healthcare (n = 53; 25.5%), and higher educational institutions (n = 110; 52.9%). Of the total participants, 95 were female, and 113 were male. In terms of job roles, 40% of the sample were managers, and 60% were workers, resulting in an even distribution across hierarchical levels. Moderate to high levels of international exposure were reported by most respondents, but about, 32%) had little to no direct overseas exposure, which supports the use of IE as a moderator in the configurational model. Overall, the respondents showed moderate to high levels of DCI, with average Likert scores ranging from 4.6 to 5.2, indicating a generally higher level of digital and intercultural competencies among GCC professionals. Table 1: Athletes' demographic and sectoral profiles.

The cross-sector, cross-country, and cross-demographic profile diversity of this sample is for the fsQCA method to find equifinal combinations of conditions across multiple cases [30]. Such heterogeneity enhances configurational validity and enables meaningful cross-sectoral comparisons in subsequent analyses.

### 3.2. Fuzzy-Set Qualitative Comparative Analysis (fsQCA): Conceptual and Methodological Foundations

Fuzzy-set Qualitative Comparative Analysis (fsQCA) is a set-theoretic analytical approach designed to examine how combinations of conditions jointly produce an outcome, rather than estimating the net effect of individual variables in isolation

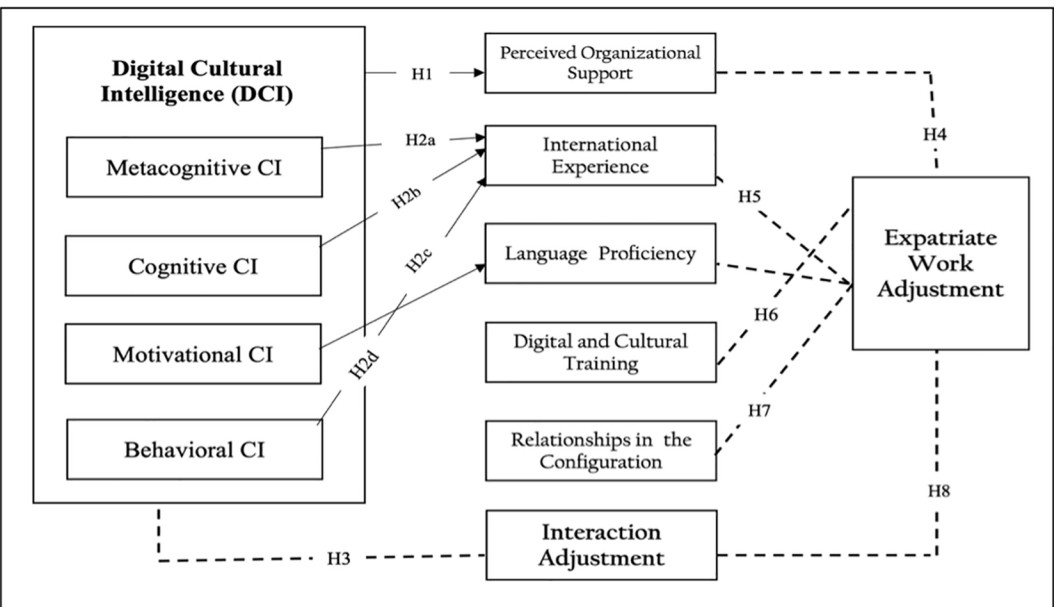

**Fig 1. Configurational framework linking Digital Cultural Intelligence dimensions to expatriate work adjustment in global work environments.**
Source: By authors.

**Table 1. Sample Distribution and Demographic Profile (N = 208).**

| Variable | Category | Frequency | Percentage (%) |
|---|---|---|---|
| **Country** | Saudi Arabia | 81 | 38.9 |
| | United Arab Emirates | 58 | 27.9 |
| | Oman | 39 | 18.8 |
| | Qatar | 30 | 14.4 |
| **Sector** | Tourism | 45 | 21.6 |
| | Healthcare | 53 | 25.5 |
| | Higher Education | 110 | 52.9 |
| **Gender** | Female | 95 | 45.7 |
| | Male | 113 | 54.3 |
| **Position** | Manager | 83 | 39.9 |
| | Employee | 125 | 60.1 |
| **International Experience** | High | 141 | 67.8 |
| | Low/None | 67 | 32.2 |

[31]. Rooted in configurational thinking, fsQCA assumes that social phenomena are characterized by causal complexity, where outcomes emerge from multiple interacting factors that may operate differently across cases.

Conceptually, fsQCA is built on three core principles. First, conjunctural causation suggests that conditions rarely exert influence independently, but instead operate in combination with other conditions. Second, equifinality recognizes that multiple, qualitatively different configurations of conditions can lead to the same outcome. Third, causal asymmetry implies that the presence and absence of an outcome are not explained by mirror-opposite causal relationships, meaning that the factors leading to successful adjustment may differ fundamentally from those associated with maladjustment [31,32].

Methodologically, fsQCA is particularly suitable for research contexts characterized by heterogeneity, limited sample sizes, and complex interaction effects—conditions that are common in expatriate research and international HRM studies. Rather than requiring assumptions of linearity, normality, or variable independence, fsQCA allows researchers to model partial membership in sets using fuzzy scores ranging between 0 and 1, thereby capturing degrees of condition presence and outcome attainment [31].

In the context of the present study, fsQCA offers a powerful analytical lens for examining EWA across diverse sectors and institutional environments within the GCC. Expatriates operating under similar regulatory and cultural conditions may nonetheless achieve successful adjustment through different combinations of DCI dimensions, POS, and experiential factors. A regression-based approach would obscure such configurational diversity by focusing on average effects, whereas fsQCA enables the identification of multiple, equally effective pathways to adjustment.

Accordingly, the use of fsQCA aligns closely with the study's theoretical emphasis on heterogeneous adaptation pathways and provides a methodologically coherent means of linking DCI to EWA in complex, digitally mediated work environments.

Although fsQCA adopts a configurational logic often associated with qualitative reasoning, the present study is based on systematically collected survey data rather than interviews or narrative accounts. Accordingly, the analysis focuses on identifying combinations of conditions leading to EWA, rather than eliciting participants' subjective narratives.

### 3.3. Sectoral scope and sample selection within the GCC context

Tourism, healthcare, and higher education were selected as focal sectors to capture the structural and functional diversity in expatriate work adjustment (EWA) across GCC countries, rather than to represent a single occupational or institutional logic. These sectors are central to national development strategies in the GCC and are characterized by substantial reliance on expatriate labor, intensive service interactions, and growing digital mediation [14,33].

GCC states have prioritized economic diversification, human capital development, and service-sector expansion in their long-term national visions, including [15,34,35,36].

Within these frameworks, tourism, healthcare, and higher education are consistently identified as growth-oriented sectors that require specialized international expertise (IE) and ongoing engagement of expatriate professionals [33].

While all three sectors depend on expatriate labor, they differ markedly in institutional logics and adjustment requirements. Tourism and hospitality are highly service-intensive, involving frequent intercultural encounters, emotional labor, and continuous interaction with diverse customers. In contrast, healthcare expatriates operate within tightly regulated professional systems, where cultural adaptation is influenced by standardized protocols, ethical norms, and digital clinical infrastructures. Higher education is a knowledge-intensive sector, with expatriate academics and administrators engaged in complex cognitive tasks, virtual collaboration, and ongoing participation in professional and organizational networks.

Regional labor market statistics indicate that expatriates comprise a substantial majority of the workforce in hospitality and healthcare services across GCC countries. Higher education institutions also remain heavily reliant on internationally recruited faculty and staff to support teaching, research, and administrative functions [33]. The pronounced differences between expatriates in higher education and those in tourism or healthcare are viewed as a theoretical advantage within a configurational research design, rather than a methodological limitation.

Fuzzy-set Qualitative Comparative Analysis (fsQCA) is well suited to this sectorally diverse context because it enables the identification of multiple equifinal pathways through which expatriates achieve successful work adjustment under varying institutional and occupational conditions. Including sectors with distinct interactional, regulatory, and digital characteristics enhances the study's ability to explain how similar adjustment outcomes can arise from different combinations of individual competencies and contextual factors across the GCC.

Data collection was conducted across several major urban centers within GCC countries to ensure contextual variation while maintaining strict confidentiality. To protect participant anonymity and comply with ethical research standards,

specific city identifiers are not disclosed. Consequently, the sample reflects geographically dispersed organizational settings rather than a single localized labor market.

### 3.4. Ethical statement

Ethical approval for this study was obtained from the Institutional Review Board (IRB) of King Khalid University, Saudi Arabia (Approval No. ECM#2025−1807, dated 06/10/2025). Before data collection, all participants were clearly informed about the academic purpose of the study, the voluntary nature of their participation, and the confidentiality and anonymity of their responses. The online questionnaire was distributed electronically, and informed consent was implied through the participants' voluntary completion and submission of the survey form. No minors were involved in this research, and all procedures strictly adhered to the ethical standards and guidelines of King Khalid University. The data collection period began on 07/10/2025 and concluded on 16/10/2025.

"Additional information regarding the ethical, cultural, and scientific considerations specific to inclusivity in global research is provided in the Supporting Information (S1 File)."

## 4. Results

### 4.1. Calibration of conditions

Based on the [31] method of direct calibration, variables were calibrated into fuzzy sets with values ranging from 0 (completely out of the set) to 1 (completely in the set) on a seven-point Likert scale. Three thresholds were set at 2.0 (outer limit of non-membership), 4.0 (crossover point), and 6.0 (full membership). The calibration was performed on eight prior conditions (four based on the dimensions of DCI Metacognitive, Cognitive, Motivational, and Behavioral) and three contextual factors (Perceived Organizational Support – POS, International Experience – IE, DCT, Language Proficiency – LP).

The final outcome, EWA was calibrated in the same way. A robustness check using percentile-based anchors (95/50/5) derived from the data showed that the membership scores were stable, with a Spearman correlation greater than 0.95 across the two calibration sets.

### 4.2. Analysis of necessary conditions

As summarized in Table 2, none of the ten conditions, even superficially, reached the threshold of Necessity of Adherence ≥ 0.90. This implies that high expatriate adjustment cannot be attributed to any single element, thereby supporting the configurational premise of conjunctural causation [37].

### 4.3. Configurational analysis for high EWA

**4.3.1. Truth table and thresholds.** Table 3 was developed with a frequency "cutoff" of 5 cases, and a consistency "cutoff" of 0.80, in accordance with [31,32]. This resulted in five sufficient solutions for high EWA. The middle solution (using directional expectations from theory) was retained for interpretive ease.

**Overall solution consistency = 0.83**, **Overall coverage = 0.80, Interpretation of Configurations.**

### C1: (METACOG● × BEH● × POS●)

Metacognitive-behavioral synergy supported by organizational facilitation is represented. Expatriates with self-reflection, adaptive communication, and POS have a high adjustment level, confirming H3 and paralleling [25] Arrangement 1.

### C2: (COG ● × MOT● × LP● × DCT● × ~IE)

This version of a model term considers cognitive understanding, motivation, and skill-based support related to having limited IE. This lends support to H4 that DCT is an adequate proxy for previous cross-cultural exposure. This leads to the hypothesis,

**Table 2. Analysis of Necessary Conditions for High EWA.**

| Condition | Consistency | Coverage | Necessity |
|---|---|---|---|
| Metacognitive DCI | 0.86 | 0.71 | No |
| Cognitive DCI | 0.83 | 0.65 | No |
| Motivational DCI | 0.84 | 0.68 | No |
| Behavioral DCI | 0.88 | 0.70 | No |
| Perceived Organizational Support | 0.87 | 0.74 | No |
| International Experience | 0.79 | 0.68 | No |
| Language Proficiency | 0.83 | 0.69 | No |
| Training (Digital–Cultural) | 0.85 | 0.72 | No |

**Table 3. Configurations Leading to High (EWA↑).**

| Configuration | METACOG | COG | MOT | BEH | POS | IE | LP | DCT | Consistency | PRI | Raw Coverage |
|---|---|---|---|---|---|---|---|---|---|---|---|
| **C1** | ● | ○ | — | ● | ● | ○ | ○ | — | **0.894** | 0.378 | 0.605 |
| **C2** | ○ | ● | ● | ○ | — | ⊗ | ● | ● | **0.924** | 0.302 | 0.315 |
| **C3** | — | ○ | ● | ● | ○ | — | — | ● | 0.866 | 0.322 | 0.604 |
| **C4** | ○ | ● | — | ● | ⊗ | — | ● | ○ | 0.864 | 0.307 | 0.579 |
| **C5** | — | ● | ○ | ● | ● | — | — | ○ | 0.879 | 0.359 | 0.609 |

As illustrated by levels C3 and C4, there are two behaviorally based channels through which high behavioral DCI with either training or POS is particularly effective in facilitating adjustment (especially in: I highly recommend operationally intense industries such as tourism and healthcare).

C5 represents cognitive-behavioral enhancement at the high end of institutional strength and suggests that even with moderate motivational intensity, contextual facilitators (POS) continue to hold primacy.

## 4.4. Configurational analysis for low (EWA↓)

To confirm causal asymmetry, a separate analysis of the negated outcome (~EWA) was performed.

The three configurations linked to low adjustment are listed in Table 4.

The negated solution confirms the asymmetry of the causal relationship [32]: lack of digital cultural competencies and POS results in low EWA, but the negation of these conditions does not imply the negation of high adjustment – high adjustment is not precluded by the absence of these conditions.

## 4.5. Robustness test

Four robustness tests were performed.

1. Alternative Calibration Anchors: The membership profiles at the 95/50/5 percentiles were similar ($\rho > 0.95$).

**Table 4. Configurations for low EWA adjustment (~EWA).**

| Configuration | METACOG | COG | MOT | BEH | POS | IE | LP | DCT | Consistency | Raw Coverage |
|---|---|---|---|---|---|---|---|---|---|---|
| **N1** | ⊗ | ⊗ | ⊗ | ⊗ | ⊗ | — | — | ⊗ | **0.892** | 0.611 |
| **N2** | — | ⊗ | ⊗ | — | ⊗ | ⊗ | — | ⊗ | 0.871 | 0.583 |
| **N3** | ⊗ | ○ | ⊗ | ⊗ | ○ | — | ⊗ | — | 0.853 | 0.562 |

2. Consistency Thresholds: The results at 0.80 and 0.85 were the same in terms of dominant configurations (C1 and C2).

3. Changing the Frequency Cutoff: Increasing the minimum frequency from five to six cases, the coverage of the solution remained the same ($0.79 \rightarrow 0.78$).

4. Out-of-Sample Predictive Validity: By splitting the data into training (n = 139) and holdout (n = 69) subsamples, the solution unity and coverage were 0.82 and 0.77, respectively, providing strong support for predictive stability [30].

## 5. Discussions

### 5.1. The configurational findings

The configurational findings suggest that none of the conditions is sufficient on its own for an outcome of high EWA. Rather, this progression of adaptation is enabled by distinct sets of digital, cultural, and contextual enablers, expressing equifinality across complex organizational systems [32].

Two configurations, C1 and C2, were identified as core paths to high adjustment, and three others were found as marginal paths: C3 to C5 contributed to secondary causal complexity but were not identical. The findings are conceptually in line with earlier configurational research [25,38] but extend this by introducing the digital facet of CI into the adaptation process.

#### 5.1.1. Interpreting configurational diversity through heterogeneous adaptation pathways.
The configurational diversity identified across the five high-adjustment solutions (C1–C5) offers robust empirical support for the proposition that expatriate work adjustment (EWA) follows heterogeneous adaptation pathways, even among individuals situated within comparable institutional and cultural contexts. Rather than implying a singular dominant logic of adjustment, the findings demonstrate that expatriates with similar backgrounds can attain successful outcomes through distinct combinations of digital-cultural competencies, organizational resources, and experiential conditions.

This pattern aligns with segmented assimilation perspectives, which argue that individuals exposed to similar structural environments do not necessarily converge toward uniform integration outcomes, but instead follow differentiated paths shaped by resource availability, social positioning, and contextual constraints [20,21]. In this study, such differentiation is evident in the contrast between configurations that rely primarily on metacognitive–behavioral capabilities combined with perceived organizational support (POS), as in C1, and those in which cognitive–motivational resources and digital communication technologies (DCT) compensate for limited international experience (IE), as in C2.

Viewed through a social network perspective, these divergent configurations reflect variations in expatriates' degrees of organizational and relational embeddedness. Expatriates who benefit from strong organizational structures and demonstrate adaptive digital communication behaviors are more likely to integrate effectively into both formal and informal work networks, thereby facilitating smoother adjustment. Conversely, when such embeddedness is weaker, alternative pathways, such as intensive training, language proficiency (LP), and motivational orientation, serve as substitutes, enabling adjustment through different relational and digital mechanisms.

This configurational evidence further supports the argument that EWA in digitally mediated environments is not determined solely by cultural background, but rather by how individuals mobilize digital-cultural intelligence (DCI) to navigate institutional boundaries and position themselves within organizational and networked contexts. The presence of multiple high-adjustment configurations highlights the value of a configurational approach, as linear models would obscure these differentiated yet equally viable adaptation trajectories.

#### 5.1.2. High-adjustment configurations.
**Configuration C1: METACOG● × BEH● × POS●:** This partnership represents an overall metacognitive–behavioral interaction hardened by active POS. Expatriates who receive institutional support and possess high levels of self-reflective capacity (i.e., metacognitive DCI) and adaptive digital behaviors will have the potential to successfully engage in intercultural virtual contexts, and such potential would be maximized by institutional support systems.

This pattern resonates with Lei et al.'s CI-driven paths, although extended to digital contexts, as at the same time it captures digital cues through reflection, promoting social attunement and virtual empathy.

Proposition 1 (P1): Expatriates with high metacognitive and behavioral DCI have best adjustment if the level of POS is high.

**Configuration C2: COG ●×MOT●×LP●×DCT●×~IE:** This configuration reflects compensatory logic: in the case of limited IE, DCT and language competence (LP) act as proxies for experiential knowledge. Motivational engagement is the underlying psychological mechanism that enables efficient digital knowledge transfer across cultures.

This finding supports the idea of resource substitution as a response to dwindling resources [39] and suggests that training and intrinsic motivation serve as replacements for experience, a particularly significant finding among younger, digitally native expatriates.

Proposition 2 (P2): The lack of IE can be complemented by a cognitive perspective, a motivational orientation, digital–cultural expertise, and LP.

**Configuration C3: MOT●×BEH●×DCT●:** This development highlights the behavioral and motivational core of the DCI, which is enhanced by focused training. It is a learning-based business adaptation route that dominates person-based industries such as tourism and healthcare. Behavioral adaptability enhances the positive effects of saliva digital–cultural training, which in turn results in faster situational adaptation in face-to-face settings.

Proposition 3 (P3): Under the condition that both motivational and behavioral DCI are high and with the addition of the training effect, expatriates will achieve a high level of adjustment even in a highly changeable service setting.

**Configuration C4: METACOG●×COG●×DCT●:** This structure reflects a cognitive−reflective DCI dimension, supported by digital training, and is more prevalent in knowledge-based organizations (e.g., universities). This implies that expatriates who practice self-reflective learning and theoretical processing of cultural knowledge may flourish in such digitally mediated academic environments.

Proposition 4 (P4): Metacognitive and cognitive DCI, together with d-t cultural training, predict adjustment in knowledge-intensive environments.

**Configuration C5: BEH●×LP●×POS●:** This pertains to behavioral flexibility with linguistic and POS as language agents of influence, underscoring the importance of communication quality in modulating digital partnerships. Linguistic ability functions as a cultural mediator, whereas PoS buffers adaptation pathways over time.

Proposition 5 (P5): Behavioral adaptability, LP, and POS serve as predictors of sustained EWA in culturally diverse d-teams.

**5.1.3. Comparative Interpretation with Low-Adjustment Configurations.** The negative combinations (N1–N3) illustrate how positive outcomes are not simply the mirror image of negative ones (i.e., while metacognition and DCI-facilitation lead to high adjustment, the concurrent absence of these factors [and further combined with organizational weakness] leads to low [mal]adjustment). This validates that reverse results are generated by different causal recipes rather than simply missing ingredients for positive recipes [39].

These asymmetrical outcomes further illustrate the contingency and interconnectedness of DCI domains: high motivation without behavioral flexibility or the environment to support will not be a recipe for succeed [9,12].

## 5.2. Theoretical Implications

**An Extension of Person-Environment Fit Theory**: The study highlights that fit in digital multicultural environments is attained not only through value congruence but also through D–C synergy, in which people harmonize their technological activities with (organizational and cultural) expectations (Kristof-Brown et al., 2005).

• DCI as a theoretical building block in configurational theory

By integrating digital competence into CI theory, this study enhances the conceptual breadth of CI to include technology-based cognition and behavior. This study expands the conceptual scope of CI to include technology-mediated cognition and behaviors.

- Progression of the Neo-Configurational HRM Discourse

The results demonstrate the two principles of equifinality and asymmetry and suggest that multiple paths, rather than linear causal chains, explain EWA in globalized workplaces.

The findings of this study advance international human resource management (HRM) scholarship by demonstrating that EWA operates as a configurational and network-embedded process within institutionally bounded environments. While the empirical context of the Gulf Cooperation Council (GCC) is unique, the theoretical implications are applicable beyond this region. Specifically, the study illustrates how DCI interacts with social embeddedness and organizational structures to generate multiple viable pathways to environmental fit. These insights are especially pertinent for scholars investigating expatriation and global work in settings marked by regulatory constraints, temporary mobility, and increasing digital mediation, even when institutional arrangements differ from those in the GCC..

### 5.3. Managerial implications

- Strategic HR Development: Organizations need to develop holistic DCI training interventions encompassing technical, linguistic, and reflective elements.

- Specific interventions: For expatriates with little international experience, motivation-based education and digital simulations can simulate the experiential impact of prior exposure.

- Enabling Factors: Organizational leaders have the responsibility of creating an environment that promotes digital empathy and agility in collaboration, especially when considering hybrid or virtual teams.

### 5.4. Methodological implications

In terms of methodology, our study demonstrates that fsQCA can be a powerful, complexity-sensitive tool in HRM research, revealing equifinal paths in an asymmetric manner that cannot be obtained by regression-based methods.

The joint use of quantitative calibration and qualitative interpretation of the configurations provides deeper insight into the interaction effects of the conditions on expatriate outcomes, which aligns with the novelties proposed by [30].

## 6. Conclusion

We are, therefore, one of the first studies to adopt a configurational approach based on the contextualized nature of DCI to investigate how it relates to the EWA under the digitally transformed work environments in the Gulf region. Adopting fsQCA approach, the study goes beyond linear models and reveals a number of equifinal solutions leading to successful adaptation of expatriates in technology-mediated, multicultural environments.

The results indicate that DCI is not a single ability but a dynamic system of intertwined and mutually supportive constructs consisting of metacognitive reflection, cultural knowledge, motivational drive, and adaptive digital behavior. These facets are not operating in a vacuum, but rather, interact with POS, IE, LP, and digital–cultural training. This interaction leads to several high adjustment configurations, thus confirming that expatriate success is contextual and achieved by a diverse range of adaptive logics rather than by a single universal solution.

The findings contribute to theory and practice. In the realm of theory, the research enhances the Person – Environment Fit Theory with the concept of Digital – Cultural Fit, under which successful adaptation occurs as a result of matching between individual and environmental digital cultural competencies. In terms of methodology, it illustrates the contribution of fsQCA in HRM studies by revealing causal asymmetry and equifinality and thus furthers a subtle insight into the success of expatriates within digitally endowed global organizations.

On an instrumental level, this study provides a global HR viewpoint. This implies that the development of expatriates' DCI by way of training, mentorship, and organisational backing can significantly improve cross-cultural performance even with minimum IE. and further add that "in-motion learning" of digital cultural and motivational agility is critical in sectors such as tourism, healthcare and higher education, as core pillars of the Gulf economies as articulated in Vision 2030.

Essentially [this research] transcends a false "technology versus culture" binary and posits an integrated framework of Digital Humanism where technology is an enabler rather than a replacement for intercultural engagement. Through demonstrating the mechanisms by which DCI empowers expatriates to convert cultural diversity into innovation and sustainable competitiveness, this research paves the way for a new digital HRM paradigm—one that acknowledges complexity, embraces configurational thinking, and redefines global talent management for the digital era.

## Supporting information

**S1 File. Inclusivity_Questionnaire.**
(DOCX)

**S2 File. fsQCA.**
(XLSX)

**S3 File. Quationnaire.**
(PDF)

## Author contributions

**Conceptualization:** Faiz Binzafrah.

**Data curation:** Osman Elsawy.

**Methodology:** Fatimah Mahdy.

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
