## [Decision Letter · Decision Letter 0]

26 Dec 2025

Dear Dr. mahdy,

Thank you for submitting your manuscript to PLOS ONE. After careful consideration, we feel that it has merit but does not fully meet PLOS ONE’s publication criteria as it currently stands. Therefore, we invite you to submit a revised version of the manuscript that addresses the points raised during the review process.

We look forward to receiving your revised manuscript.

Kind regards,

Dafeng Xu

Academic Editor

PLOS One

Journal Requirements:

the Deanship of Research and Graduate Studies at King Khalid University, KSA, for funding this work through .(General Research Project under grant number (GRP/5/46) -2025

the Deanship of Research and Graduate Studies at King Khalid University, KSA, for funding this work through .(General Research Project under grant number (GRP/5/46) -2025.

the Deanship of Research and Graduate Studies at King Khalid University, KSA, for funding this work through .(General Research Project under grant number (GRP/5/46) -2025

6. Please amend your authorship list in your manuscript file to include author fatimah mahdy, Faiz Binzafrah, Osman Elsawy.

7. Please remove your figures from within your manuscript file, leaving only the individual TIFF/EPS image files, uploaded separately. These will be automatically included in the reviewers’ PDF.

8. We note you have included a table to which you do not refer in the text of your manuscript. Please ensure that you refer to Table 3 in your text; if accepted, production will need this reference to link the reader to the Table.

Additional Editor Comments :

We now have two reviewers' reports. We cannot accept the paper in its current form but encourage you to revise the paper. Both reviewers mention that it is necessary to improve the literature review, and to present a better introductory section of GCC's specific contexts. We believe that the paper requires a careful round of revision, but the revision is doable.

Reviewers' comments:

Reviewer's Responses to Questions

**Comments to the Author**

1. Is the manuscript technically sound, and do the data support the conclusions?

Reviewer #1: Partly

Reviewer #2: No

2. Has the statistical analysis been performed appropriately and rigorously?

Reviewer #1: N/A

Reviewer #2: Yes

3. Have the authors made all data underlying the findings in their manuscript fully available?

Reviewer #1: Yes

Reviewer #2: Yes

4. Is the manuscript presented in an intelligible fashion and written in standard English?

Reviewer #1: No

Reviewer #2: No

Reviewer #1: This paper studies how DCI influences expatriate work adjustment in GCC countries. The paper has a theoretical framework alongside a qualitative analysis. The paper studies an interesting question, but there is also huge room for improvement. I here suggest several thoughts:

1. This paper lacks serious engagement in GCC's specific context. I believe that both the authors and readers would agree that the labor markets in GCC countries are very different from those in other countries like the United States or Europe. While the authors are welcome to do a comparative analysis, it is extremely important to first describe how GCC's specific context presents particular interests to the general audience -- what can we learn from an analysis in GCC countries in the first place?

2. A particular feature of GCC HRM and difference between GCC HRM and HRM in other countries, in my opinion, is not diversity of different cultures; rather, it is how people -- workers, managers, consumers -- from the same culture have different behavioral responses to the inter-cultural interactions, local labor markets and businesses, etc. To me, this is something I can feel in GCC countries but is less studied in the prior literature; in other countries such as the United States, there are some previous findings but they are not as relevant as those in the GCC context, which I believe is quite unique. Can the authors shed light on this point and delve into it? For example, what if two expatriates with the same cultural (or demographic) background develop different paths of "cultural fit" in a GCC country? How does this phenomenon inform policymaking or HRM? Are there any convergence or divergence of "cultural fit" based on social network behaviors?

Here are some papers the authors can take a look and consider -- they are not in the GCC context, and thus the authors may use their analysis to argue why the GCC context is particularly interesting.

Segmented assimilation: Issues, controversies, and recent research on the new second generation - M Zhou - International migration review, 1997

The social context of assimilation: Testing implications of segmented assimilation theory - Y Xie, E Greenman - Social science research, 2011

Acculturational homophily - D Xu - Economics of Education Review, 2017

Understanding inclusion - KA Brix, OA Lee, SG Stalla - BioScience, 2022

Champions of Diversity Governance? City Approaches to Cultural Heterogeneity in Europe, Japan, and South Korea - B Peruzzi Castellani - 2025 - working paper

3. The authors should present a more comprehensive introduction to sample selection. For example, even to me it is not entirely straightforward why the authors choose tourism, healthcare, and higher education as the three sectors studied in the qualitative analysis. Is it possible to provide some descriptive statistics of the representativeness of these sectors, especially among the expatriates? This is important, because expatriates working in higher education are supposed to be very different from their counterparts in hospitality, no? Also, is it possible to visualize the fieldwork sites possibly at the city level, especially for large cities where interviewees are less likely to be identified?

4. The authors should present an introduction to fsQCA, both methodologically and conceptually.

Reviewer #2: I was invited to review this paper. While this paper concerns an interesting research question, I have several serious concerns about the paper's methodological and writing issues, and I believe that this paper cannot be accepted until a round of careful revision.

My specific comments:

1. Should introduce the economic and policy contexts of Gulf Cooperation Council countries. Readers outside of Middle East might not very little about this region and its labor market.

2. I am surprised to see a paper discussing "environment fit" and "cultural fit" does not review the literature on social networks. There are many previous studies of cultural and demographic social networks, and how these networks have similar and different features. I also suggest the authors of this paper conduct a thorough literature review on previous empirical studies of the relevant topics (e.g., social networks as mentioned above). Currently, most papers the authors cite are regarding the general theoretical framework. But this paper is not a theoretical one.

3. There are many writing issues. I am not talking about grammar -- although the authors of the paper should perhaps check this as well. But more seriously, the authors should focus on two types of revisions regarding writing issues: (1) the use of abbreviations -- it is fine to use "GCC" or "UAE", and perhaps DCI as well, but there are many abbreviations that are not even pre-defined in this paper, such as EWA and HRM; (2) from Section 5 (and actually as early as Section 4.5), the paper gradually turns to be a bullet-point-type presentation rather than an academic paper; see an example of Section 5.5 -- they are purely bullet points rather than a well-organized paragraph.

4. Since it is a qualitative paper, it would be great if the readers can see some records of actual conversations, interviews, etc. of the surveyed individuals. It is great to see some illustrative results and statistics, but it would be more interesting to see people's original words.

5. I would actually be hesitant to try to frame this paper as a "global HR" paper. It seems that Gulf Cooperation Council countries are very unique, so unless the authors want to conduct a comparative analysis -- which, by the way, is certainly a way to improve this paper -- the authors of the paper should engage more in describing why this paper's unique context is interesting and important to the world. It needs not to be generalized in the practice sense, although the findings of the paper might contribute to the theoretical side (such as the network literature suggested above).

**Do you want your identity to be public for this peer review?** For information about this choice, including consent withdrawal, please see our Privacy Policy

Reviewer #1: No

Reviewer #2: No

---

## [Author Response · Author response to Decision Letter 1]

23 Jan 2026

The first reviewer’s Comment

Comment 1: The manuscript does not sufficiently explain why the GCC context is theoretically and empirically important, given the uniqueness of GCC labor markets.

Response: We sincerely thank the reviewer for highlighting the need for a stronger contextual grounding of the Gulf Cooperation Council (GCC) setting. We fully agree that the GCC labor market possesses distinctive institutional, demographic, and regulatory characteristics that warrant explicit theoretical engagement rather than being treated merely as an empirical backdrop.

In response, we have substantially revised the Introduction to include a dedicated contextual subsection that theorizes the GCC as a generative research context for expatriate work adjustment and digital cultural intelligence. Specifically, we now elaborate on (a) the structurally expatriate-dependent nature of GCC labor markets, (b) the temporariness and regulatory embeddedness of expatriate employment, and (c) the coexistence of rapid digital transformation with strong institutional and cultural boundaries. These features collectively intensify the salience of digital–cultural fit, rendering the GCC context theoretically distinctive for configurational HRM research.

This revision clarifies why findings derived from the GCC are not merely context-bound but instead offer theoretically transferable insights into expatriate adjustment under highly institutionalized, digitally mediated global work environments.

Location of revision:

Introduction – newly added subsection on the GCC labor market context.

Comment 2: The reviewer suggests shifting the focus from “diversity of cultures” to heterogeneous adaptation pathways among expatriates from similar backgrounds and encourages engagement with segmented assimilation and social network theories.

Response: We sincerely thank the reviewer for this insightful and theoretically valuable suggestion. We fully agree that expatriate adjustment should not be conceptualized solely as a function of cross-cultural diversity, but rather as a process characterized by heterogeneous adaptation trajectories, even among expatriates with similar cultural or demographic backgrounds.

In response, we have refined the theoretical framing of the manuscript to explicitly emphasize intra-group heterogeneity in expatriate adjustment. Drawing on segmented assimilation theory and social network perspectives, we now argue that expatriates originating from similar cultural contexts may follow divergent adjustment pathways depending on their digital-cultural competencies, organizational embeddedness, and network positioning within the host environment. This refinement aligns closely with the configurational logic of the study and strengthens the theoretical justification for employing fsQCA to capture multiple equifinal paths to successful adjustment.

We have incorporated this perspective into the theoretical framework and discussion sections, thereby shifting the analytical emphasis from cultural difference per se to the interaction between individual capabilities, social embeddedness, and institutional context.

Location of revision:

Section 2 – Theoretical and Conceptual Framework

Comment 3 : The reviewer requests a more comprehensive justification of sample selection, particularly the choice of tourism, healthcare, and higher education sectors. The reviewer also asks whether descriptive evidence can be provided to demonstrate the representativeness of these sectors among expatriates, and whether fieldwork sites can be visualized at the city level while maintaining anonymity.

Response: We thank the reviewer for this constructive comment regarding sample selection and sectoral scope. We agree that clearer justification strengthens methodological transparency and interpretive rigor.

In response, we have expanded the Methodology section to provide a more systematic rationale for selecting tourism, healthcare, and higher education as the focal sectors. These sectors were deliberately chosen to capture structural and functional heterogeneity in expatriate work contexts within the GCC, particularly with respect to (a) degrees of service interaction, (b) knowledge intensity, (c) digital mediation of work, and (d) exposure to culturally embedded stakeholder interactions.

We have also added descriptive contextual statistics, drawing on regional labor market reports, to demonstrate the centrality of expatriate employment in these sectors across GCC economies. Rather than assuming homogeneity, we explicitly acknowledge and theorize sectoral differences in expatriate roles and adjustment demands, which further justify their inclusion within a configurational research design.

Regarding the visualization of fieldwork sites, we respectfully note that city-level mapping was constrained by ethical and confidentiality considerations. However, we have clarified the multi-city nature of data collection and the logic of geographic dispersion in the revised manuscript, while maintaining strict anonymity protections consistent with ethical research standards.

Location of revision:

Methodology – Sample Selection and Research Context.

Comment 4: The authors should present an introduction to fsQCA, both methodologically and conceptually.

Response: We thank the reviewer for this important suggestion. We agree that providing a clearer methodological and conceptual introduction to fuzzy-set Qualitative Comparative Analysis (fsQCA) enhances transparency and improves accessibility for readers who may be less familiar with configurational approaches.

In response, we have added a dedicated subsection in the Methodology section that introduces fsQCA both conceptually and methodologically. This subsection explains the core principles of fsQCA—namely configurational causation, equifinality, and causal asymmetry—and clarifies why fsQCA is particularly well suited for examining expatriate work adjustment in heterogeneous and institutionally complex contexts such as the GCC. We also contrast fsQCA with conventional regression-based approaches to highlight its analytical advantages for the present research objectives.

Location of revision:

Methodology – Analytical Strategy (new subsection introducing fsQCA).

The second Reviewer

Comment1 : The authors should introduce the economic and policy contexts of Gulf Cooperation Council countries, as readers outside the Middle East may have limited familiarity with the region and its labor market.

Response: We thank the reviewer for highlighting the importance of contextualizing the GCC for an international readership. We fully agree that a clear explanation of the region’s economic and labor market characteristics is essential for situating the study.

In response, we have substantially expanded the Introduction to include a dedicated discussion of the GCC economic and policy context. The revised manuscript now explains the structural dependence of GCC labor markets on expatriate employment, the institutional features of contract-based and regulated mobility systems, and the distinctive policy environment shaped by national development visions across the region. We further clarify how these characteristics differentiate the GCC from traditional Western expatriation contexts and why they render the region theoretically significant for studying expatriate work adjustment and digital cultural intelligence.

This contextual addition is designed to orient readers unfamiliar with the Middle East while directly supporting the study’s theoretical framing and research design.

Location of revision:

Introduction – expanded subsection on the GCC economic and labor market context.

Comment 2: The reviewer notes that despite discussing “environment fit” and “cultural fit,” the manuscript does not sufficiently engage with the literature on social networks. The reviewer suggests incorporating prior empirical studies on cultural and demographic social networks and clarifying how this literature informs the study, given that the paper is empirical rather than purely theoretical.

Response: We thank the reviewer for this important observation and fully agree that the literature on social networks is highly relevant to understanding environment fit and cultural fit in expatriate contexts. In response, we have substantially strengthened the literature review by integrating key empirical studies on social networks, acculturational homophily, and relational embeddedness, and by explicitly linking these perspectives to expatriate work adjustment.

Specifically, we now draw on empirical research demonstrating how social and professional networks shape access to information, support, and opportunities for cultural learning, thereby influencing adjustment outcomes. Rather than treating social networks as an additional variable, we position them as a contextual mechanism through which digital cultural intelligence facilitates alignment between individuals and their work environments. This integration allows us to ground the conceptual discussion of environment fit and cultural fit in well-established empirical findings, while remaining consistent with the study’s configurational design.

These revisions ensure that the manuscript engages more directly with relevant empirical literature and clarifies how social network perspectives inform the interpretation of the results.

Location of revision:

Literature Review – expanded subsection on social networks and expatriate adjustment; Discussion section.

Comment 3: The reviewer notes several writing issues related to (1) the inconsistent use of abbreviations that are not clearly defined (e.g., EWA, HRM), and (2) the presentation style in later sections of the manuscript, where the writing shifts toward bullet-point formats rather than coherent academic prose.

Response: We thank the reviewer for this helpful comment regarding clarity and presentation. In response, we have undertaken a comprehensive editorial revision of the manuscript.

First, all abbreviations are now clearly defined at their first occurrence in the text (e.g., expatriate work adjustment [EWA], human resource management [HRM], digital cultural intelligence [DCI], and Gulf Cooperation Council [GCC]), and their usage has been standardized throughout the manuscript.

Second, we have revised Sections 4.5 through 5.5 to replace bullet-point-style presentations with fully developed academic paragraphs. These sections now present the results and discussion in a coherent narrative form, with improved transitions and clearer argumentative flow.

Finally, the manuscript has undergone careful language and style editing to ensure consistency, readability, and adherence to academic writing conventions.

Location of revision:

Throughout the manuscript; particularly Sections 4.5–5.5.

Comment 4: Since it is a qualitative paper, the reviewer suggests including records of interviews or participants’ original words to enrich the presentation of findings.

Response: We thank the reviewer for this thoughtful suggestion and appreciate the interest in gaining deeper insight into participants’ experiences. We would like to respectfully clarify that the present study is not a qualitative interview-based investigation, but rather a configurational empirical study employing fuzzy-set Qualitative Comparative Analysis (fsQCA) based on calibrated survey data.

fsQCA is a set-theoretic analytical approach that differs fundamentally from qualitative interview methods. While it adopts configurational logic, it relies on systematically collected quantitative indicators rather than narrative or conversational data. Consequently, the inclusion of interview excerpts or participants’ verbatim statements would not be methodologically consistent with the study design or data collection strategy.

To address potential ambiguity, we have clarified the methodological positioning of the study in the Methodology section, explicitly distinguishing fsQCA from qualitative interview-based research. We have also strengthened the interpretive discussion of the configurational results to provide richer, theoretically grounded explanations of expatriate experiences without implying qualitative data collection.

Location of revision:

Methodology – clarification of research design and analytical approach.

Comment 5: The reviewer expresses hesitation in framing the manuscript as a “global HR” paper, noting the uniqueness of the GCC context. The reviewer suggests that unless a comparative design is adopted, the authors should better articulate why the GCC context is globally relevant, emphasizing theoretical rather than practical generalization.

Response: We thank the reviewer for this thoughtful and constructive comment. We agree that the GCC represents a highly distinctive institutional and labor market context, and that the study does not aim to offer direct practical generalizations across all global HR settings.

In response, we have refined the framing of the manuscript to clarify that its contribution lies primarily in theoretical generalization rather than universal practical applicability. Specifically, we now position the GCC as a theoretically generative context through which broader insights into expatriate adjustment, digital cultural intelligence, and networked forms of environment fit can be developed. Rather than presenting the study as a comparative global HR analysis, we explicitly emphasize that the findings contribute to international HRM theory by illustrating how adjustment processes unfold under conditions of strong institutional constraint, high expatriate dependence, and digital mediation.

We have revised the Introduction and Discussion sections to articulate why the GCC context is analytically valuable for global audiences, while acknowledging its contextual specificity. This reframing ensures conceptual clarity and aligns the manuscript with the reviewer’s recommendation.

Location of revision:

Introduction (theoretical positioning); Discussion (theoretical implications).

---

## [Decision Letter · Decision Letter 1]

27 Jan 2026

Digital Cultural Intelligence and Its Role in Enhancing Expatriate Work Adjustment: A Configurational Approach in Global Work Environments

PONE-D-25-57282R1

Dear Dr. mahdy,

We’re pleased to inform you that your manuscript has been judged scientifically suitable for publication and will be formally accepted for publication once it meets all outstanding technical requirements.

Kind regards,

Dafeng Xu

Academic Editor

PLOS One

Additional Editor Comments (optional):

Reviewers' comments:

Reviewer's Responses to Questions

**Comments to the Author**

Reviewer #1: All comments have been addressed

2. Is the manuscript technically sound, and do the data support the conclusions?

Reviewer #1: Yes

3. Has the statistical analysis been performed appropriately and rigorously?

Reviewer #1: Yes

4. Have the authors made all data underlying the findings in their manuscript fully available?

Reviewer #1: Yes

5. Is the manuscript presented in an intelligible fashion and written in standard English?

Reviewer #1: Yes

Reviewer #1: (No Response)

**Do you want your identity to be public for this peer review?** For information about this choice, including consent withdrawal, please see our Privacy Policy

Reviewer #1: No

---

## [Editor Report · Acceptance letter]

PONE-D-25-57282R1

PLOS One

Dear Dr. Mahdy,

I'm pleased to inform you that your manuscript has been deemed suitable for publication in PLOS One. Congratulations! Your manuscript is now being handed over to our production team.

Kind regards,

on behalf of

Dr. Dafeng Xu

Academic Editor

PLOS One